PLA2G16 expression predicts prognosis and gemcitabine sensitivity in patients with pancreatic cancer

Sun Xiaoyu 1 2
Jiang Haiyang 1
Huang Yufei 1
Xia Jing 1
Gu Jie 1
Sui Xinbing 1 2
Sun Xueni 1 2 xnsun@hznu.edu.cn
Zhou Yucheng 2 drzhouyc@163.com
1 School of Pharmacy, Hangzhou Normal University , Hangzhou, Zhejiang , China
2 Department of Gastrointestinal & Pancreatic Surgery, Key Laboratory of Gastroenterology of Zhejiang Province, Zhejiang Provincial People’s Hospital , Hangzhou, Zhejiang , China
Oliveira Sonia
Electronic publication date: 2025 May 30
Publication date: 2025
Volume: 13
Electronic Location ID: e19517
Received 2024 Nov 12; Accepted 2025 May 2
Copyright: © 2025 Sun et al.
Copyright year: 2025
Copyright holder: Sun et al.
License: This is an open access article distributed under the terms of the Creative Commons Attribution License, which permits unrestricted use, distribution, reproduction and adaptation in any medium and for any purpose provided that it is properly attributed. For attribution, the original author(s), title, publication source (PeerJ) and either DOI or URL of the article must be cited.
License URL: https://creativecommons.org/licenses/by/4.0/

Keywords: PLA2G16, Pancreatic cancer, Prognosis prediction, Gemcitabine sensitivity, Immune infiltration

Funding: China Postdoctoral Science Foundation Funded Project 2023M733163 Postdoctoral Research Startup Foundation of Zhejiang Provincial People’s Hospital This work was supported by the China Postdoctoral Science Foundation Funded Project (2023M733163) and Postdoctoral Research Startup Foundation of Zhejiang Provincial People’s Hospital. The funders had no role in study design, data collection and analysis, decision to publish, or preparation of the manuscript.

==============================
Background

Pancreatic cancer is highly aggressive with limited treatment options. PLA2G16 has been implicated in cancer progression, but its role in prognosis and gemcitabine sensitivity in pancreatic cancer remains poorly understood.

Methods

Using TCGA data, the study assessed the correlation between PLA2G16 expression and patient survival. The expression of PLA2G16 in gemcitabine-resistant versus sensitive pancreatic cancer cells was also compared. siRNA-mediated knockdown of PLA2G16 was performed in drug-resistant PANC-1 cells to evaluate its impact on gemcitabine sensitivity. The relationship between PLA2G16 expression, immune infiltration, and molecular pathways in pancreatic cancer was explored using CIBERSORT and DAVID tools.

Results

PLA2G16 was significantly overexpressed in pancreatic cancer tissues and associated with poorer patient survival. In PANC-1 cells, increased PLA2G16 expression correlated with gemcitabine resistance, and its knockdown improved drug sensitivity. PLA2G16 expression was linked to specific immune infiltration patterns and cancer-related molecular pathways.

Conclusions

Elevated PLA2G16 expression is associated with poor survival and gemcitabine resistance in pancreatic cancer, making it a potential prognostic marker and therapeutic target.

Introduction

Pancreatic cancer stands as a formidable and recalcitrant disease, constituting one of the leading causes of cancer-related mortality (Sung et al., 2021). It is frequently diagnosed at advanced stages when it has already spread to surrounding organs or distant sites, making treatment more difficult and reducing the chances of successful outcomes (Kleeff et al., 2016). The clinical treatment of pancreatic cancer depends on many factors, currently including the size of the tumor, the location of the disease, the stage of the disease, and the physical condition of the patient (Conroy et al., 2023; Stoop et al., 2024). Surgical resection is the best treatment for localized tumors, where the aim is to remove the tumor and surrounding involved tissue (Kleeff et al., 2007). Chemotherapy is often administered in combination with surgery or radiation therapy. The most commonly used chemotherapy drug for pancreatic cancer is gemcitabine, either alone or in combination with other agents such as nab-paclitaxel (Burris et al., 2023; Philip et al., 2020; Von Hoff et al., 2013), albeit yielding palliative benefits and modest survival extensions rather than definitive cures. The high potential for metastasis, late-stage diagnosis, and resistance to conventional therapies contribute to the limited success rates. Consequently, there is an imperative to foster the development of individualized and targeted therapeutic strategies, particularly by identifying molecular targets that can enhance patients’ sensitivity to chemotherapy (Brozos-Vázquez et al., 2024).

PLA2G16 (also referred to as HREV107, HRSL3, HRASLS3, PLAAT3, and AdPLA et al.) represents a recent addition to the phospholipase A2 (PLA2) family. PLA2G16 assumes a pivotal physiological role in lipolysis and metabolism (Jaworski et al., 2009). Prior investigations have demonstrated a correlation between aberrant PLA2G16 expression in certain tumor cells and unfavorable prognosis, yet the precise molecular mechanisms underlying tumor development remain incompletely understood (Ding et al., 2020; Roder, Latasa & Sul, 2002; Yang et al., 2017). PLA2G16 present high expression, non-small cell lung cancer (NSCLC) patients are often present a poor prognosis (Nazarenko et al., 2006). PLA2G16 overexpression promotes the proliferation and metastatic potential of some osteosarcoma cell lines and patients (Cheng et al., 2019; Liang et al., 2015). Additionally, PLA2G16 plays crucial roles in sustaining breast cancer stem cell (BCSC) properties and can be used as a biomarker for BCSCs or other cancer stem cells (Liu et al., 2021). Therefore, PLA2G16 may have oncogenic roles in specific human tumors. PLA2G16 is associated with the prognosis of patients with pancreatic cancer (Xia et al., 2020). Up-regulation of its expression can promote glycolysis and promote the growth of pancreatic cancer cells (Xia et al., 2020). Thus, further elucidation of the molecular mechanisms governing PLA2G16 in tumor initiation and progression, alongside a comprehensive understanding of its involvement in tumor cell proliferation, holds promising potential for its application as a novel tumor biomarker and therapeutic target in clinical settings, including pancreatic cancer.

This investigation examined the association between PLA2G16 expression levels and prognostic outcomes in individuals with pancreatic cancer, with particular emphasis on the responsiveness of pancreatic cancer cells to gemcitabine. PLA2G16 presents itself as a potential target for therapeutic interventions in pancreatic cancer management, particularly in mitigating resistance to gemcitabine-based treatments. Furthermore, notable correlations were identified between PLA2G16 expression levels and the infiltration of immune cells in pancreatic cancer. PLA2G16 may serve as both a prognostic marker and therapeutic target in pancreatic cancer.

Materials and Methods

All reagents and experimental instruments used in this study are listed in Tables S1 and S2, respectively.

Data acquisition

To investigate the impact of PLA2G16 expression on pancreatic cancer patients, we acquired a pancreatic cancer patient dataset comprising clinical information and gene expression profiles from the publicly accessible repository, The Cancer Genome Atlas (TCGA) (Blum, Wang & Zenklusen, 2018). Specifically, we focused on tumor tissues that received gemcitabine treatment, and any missing cases were appropriately handled. The RNA sequencing data underwent a transformation process to convert the gene expression values to a format that closely approximated microarray results, enabling compatibility and comparability in subsequent analyses. According to the expression level of PLA2G16, tumor tissues were divided into high expression group and low expression group to evaluate the effect of PLA2G16 expression on tumors. To complement our study, SW1990 cell samples after gemcitabine treatment were retrieved from the Gene Expression Omnibus (GEO) database. The dataset (GSE172303) comprised cell lines exhibiting sensitivity to gemcitabine and cell lines demonstrating resistance to gemcitabine. We specifically focused on screening the expression levels of PLA2G16 in relation to the cell line samples.

Correlation analysis between overall survival and PLA2G16 expression in patients with pancreatic cancer

To study the relationship between the expression of PLA2G16 and clinicopathological features in pancreatic cancer, we employed GEPIA, an online tool widely utilized for gene expression analysis in cancer research (Tang et al., 2017). By leveraging GEPIA, we conducted survival curve analysis based on differential PLA2G16 expression to examine its correlation with the prognosis of pancreatic cancer patients.

Gene differential expression and functional enrichment analysis

Using RStudio, differentially expressed genes (DEGs) between PLA2G16 high expression and low expression group were identified and subsequently visualized through the generation of a volcano plot, which displays the significance and fold change of gene expression differences between the groups. Additionally, a heat map was constructed to provide a comprehensive overview of the expression patterns of the top DEGs, allowing for the visualization of potential gene expression signatures associated with PLA2G16 expression levels in pancreatic cancer. To further understand the biological processes and pathways associated with DEGs between the PLA2G16 high and low expression groups, Gene Ontology (GO) and Kyoto Encyclopedia of Genes and Genomes (KEGG) enrichment analyses were performed using the DAVID website. Terms in GO and KEGG with an adj.P.Val below 0.05 were considered significantly enriched. To visualize and interpret the enriched terms from the GO and KEGG analyses, a bioinformatics website (https://www.bioinformatics.com.cn) was employed.

Cell culture

The human pancreatic cancer cell lines PANC-1 and BxPC-3 were obtained from Sun Yat-sen University and the Cell Bank of the Chinese Academy of Sciences (Shanghai, China), respectively. PANC-1/GR and PANC-1 cells were cultured in DMEM medium (Cat# L110KJ; Shanghai Yuanpei), while BxPC-3/GR and BxPC-3 cells were maintained in RPMI-1640 medium (Cat# L210KJ, Shanghai Yuanpei). Both media were supplemented with 10% fetal bovine serum and 1% penicillin-streptomycin. For resistant cell lines (PANC-1/GR and BxPC-3/GR), 5 μM gemcitabine was added to maintain drug resistance. All cells were incubated at 37 °C in a humidified 5% CO₂ atmosphere.

Construction of drug-resistant cells

BxPC-3 and PANC-1 cells within 5–10 passages after thawing to minimize genetic drift were treated with gemcitabine (Cat#HY-17026; Med Chem Express, Monmouth Junction, NJ, USA) at an initial concentration of 0.50 μM (1/4 IC50 for BXPC-3) or 0.49 μM (1/4 IC50 for PANC-1). The medium was replaced every 48 h with fresh medium containing the same concentration of gemcitabine. The gemcitabine concentration was incrementally increased by 0.50 μM every 1–2 weeks until reaching 50 μM. Drug-resistant cells (BxPC-3/GR and PANC-1/GR) were continuously cultured for 3 months under these conditions.

CCK-8 assays

Cell viability was assessed using the Cell Counting Kit-8 (Meilunbio, Dalian, China). PANC-1/GR cells were cultured in complete DMEM medium (10% FBS, 1% penicillin/streptomycin) at 37 °C/5% CO₂ until 70–80% confluent, then harvested using 0.25% trypsin, neutralized with complete medium, and centrifuged to obtain a cell pellet. After resuspension with cell culture medium and counting, cells were adjusted to 5,000 cells/100 μL in complete medium and seeded into 96-well plates (100 μL/well), followed by overnight incubation to allow adhesion. To calculate the resistance index of different cell lines, cells were then treated with gemcitabine at concentrations of 0, 4, 8, 16, 32, 64, and 128 μM for 48 h. Subsequently, 10% CCK-8 reagent was added to each well, and plates were incubated for 2 h at 37 °C. Absorbance at 450 nm was measured using a multifunctional microplate reader. Cell viability (%) is calculated by normalizing absorbance values against untreated controls, and IC50 is derived by fitting dose-response data to a sigmoidal curve using GraphPad Prism 8.0.

Western blot analysis

Cells were cultured under standard conditions. Upon reaching a confluency of 80%, the cells were harvested, and protein extraction was performed. Subsequently, Western blot analysis was carried out. Cells were collected to prepare whole-cell lysates using cell lysis buffer supplemented with protease and phosphatase inhibitors. The protein content was measured using BCA Protein Assay Kit. Then, 20 µg of protein was separated by SDS-PAGE and transferred to PVDF membrane. By continuous shaking, the membrane was then incubated in 5% skim milk for an hour at room temperature. The membranes were then incubated with the anti-PLA2G16 antibody (Cat#A16018; Abclonal, Woburn, MA, USA) and anti-β-actin antibody (Cat#AC038; Abclonal, Woburn, MA, USA) overnight at 4 °C, followed by a PBST wash (3 × 10 min) and incubation with the appropriate secondary antibody for 2 h at room temperature. After that, the membrane was again washed with PBST (3 × 10 min) on a shaker. Finally, the protein bands were recorded using ECL detection reagents.

RT-qPCR analysis

Total RNA was extracted from cultured cell lines using TRIzol reagent (Precision Biology, Durham, NC, USA), and concentrations were determined using a NanoDrop ND-2000 spectrophotometer (Thermo Fisher Scientific, Waltham, MA, USA). The measured concentrations of total RNA were reverse transcribed into cDNA using 4× gDNA wiper Mix and 5× HiScript III qRT SuperMix (Vazyme, Beijing, China). 2× ChamQ Universal SYBR (Vazyme, Beijing, China), CFX96 Touch (Bio-Rad, Hercules, CA, USA) were used for PCR detection. Assay results were normalized to GAPDH expression and calculated using the 2−∆∆CT method. The primers used were as follows:

human PLA2G16, 5′-CCAGGTCAACAACAAA-CATGATG-3′ (forward) and 5′-CCCGCTGGATGATTTTGC-3′ (reverse);

human GAPDH, 5′-TGATGACATCAAGAAGGTGGTGAAG-3′ (forward) and 5′-TCCTTGGAGGCCATGT-GGGCCAT-3′ (reverse).

siRNA transfection

The targets sequences of siPLA2G16 were GGAGUCAUGUUCUCAAGAATT (siPLA2G16-1) and GCGAGCACUUUGUGAAUGATT (siPLA2G16-2). Negative controls and siRNAs were synthesized by GenePharma (Shanghai, China). Transient transfection was performed using Lipo2000 reagent (Vazyme, Beijing, China). RT-qPCR was conducted to detect transfection efficiency.

Colony formation assays

For the colony formation assay, approximately 4 × 10³ PANC-1/GR cells per well were seeded in 6-cm cell culture dishes and maintained for approximately 15 days. Following incubation, cells were washed 2–3 times with PBS and fixed with 4% paraformaldehyde. After removing the fixative with PBS, the cells were stained with crystal violet. Images were captured using an appropriate camera, and colonies were quantified using ImageJ. The resulting data were subsequently analyzed using GraphPad Prism 8.0 for statistics analysis.

Wound-healing assay

Approximately 7 × 10⁵ PANC-1/GR cells per well were seeded in 6-well plates and cultured until the cell density exceeded 90%. A 200 μL pipette tip was then used to scratch the bottom of the wells, creating a cell-free area. The cells were subsequently cultured for 24, 48, and 72 h. Scratch area was monitored using a photographic microscope (Nikon, Tokyo, Japan), and scratch width was quantified using ImageJ software.

Genetic mutation analysis

The potential involvement of PLA2G16 mutations was investigated using data extracted from the TCGA dataset through CBioPortal (https://www.cbioportal.org/). The mutation and structural variant analysis of the PLA2G16 gene was conducted utilizing the “OncoPrint” module available on the cBioPortal platform.

Immune infiltration analysis

To evaluate the immune response and assess its correlation with survival and molecular subpopulations in pancreatic cancer, we employed CIBERSORT (Gentles et al., 2015), a deconvolution algorithm based on gene expression analysis. By evaluating gene expression changes relative to the entire gene set, CIBERSORT is able to accurately estimate the frequency of 22 tumor immune-infiltrating cell types (TIICs) in a sample.

Statistical analysis

The data were processed and analyzed using GraphPad Prism 8.0. Cell viability was evaluated using five biological replicates per group. Data for the colony formation and wound-healing assays were derived from three independent experiments. Statistical differences between two groups with a normal distribution were assessed using Student’s t-test. For comparisons involving more than two groups with a normal distribution, analysis of variance (ANOVA) was performed, followed by Dunnett’s multiple comparisons test. A p-value of less than 0.05 was considered statistically significant.

Results

PLA2G16 is highly expressed in pancreatic cancer

Figure 1A illustrates the expression of PLA2G16 in pancreatic cancer, as extracted from GEPIA, revealing a significant elevation in PLA2G16 expression within tumor tissues compared to normal tissues. Additionally, PLA2G16 protein expression was evaluated utilizing immunohistochemistry (IHC) staining data acquired from the Human Protein Atlas (HPA). In normal pancreatic tissue, PLA2G16 expression predominantly remained low in exocrine glandular cells (depicted in Fig. 1B, left). Conversely, among 10 cases of pancreatic adenocarcinoma (PAAD) in the HPA dataset, four cases exhibited high or medium expression of PLA2G16 (as illustrated in Fig. 1B, right). These cumulative findings corroborate the consistently heightened expression of PLA2G16 in individuals diagnosed with pancreatic cancer, underscoring the potential significant role of PLA2G16 in the pathogenesis of pancreatic cancer.

Figure 1 PLA2G16 is highly expressed in pancreatic cancer.

(A) Differential expression of PLA2G16 in pancreatic cancer tissues (n = 179) and normal controls (n = 171), *: p < 0.05. (B) Representative images of IHC staining of PLA2G16 in normal pancreas and PAAD tissues (data were obtained from the HPA database). (C–E) Mutation status analysis of PLA2G16 in pancreatic cancer.

Additionally, genetic mutations are significant contributors to tumorigenesis and prognosis. To explore the potential impact of PLA2G16 mutations in pancreatic cancer, we examined the mutation frequency and structural variants of the PLA2G16 gene using data obtained from the cBioPortal database, encompassing six studies and 1,174 samples. As depicted in Figs. 1C–1E, the mutation frequency of PLA2G16 in pancreatic cancer appears relatively low, at 0.3%, with amplification identified as the most prevalent genetic alteration, followed by deep deletion.

The expression of PLA2G16 strongly correlates with the OS of pancreatic cancer patients

We then conducted an analysis to investigate the association between PLA2G16 expression and the survival rate of patients with pancreatic cancer, aiming to elucidate the significant role of PLA2G16 in this disease. Clinical data from 178 patients diagnosed with pancreatic ductal adenocarcinoma (PDAC) sourced from The Cancer Genome Atlas (TCGA) are summarized in Table 1. Then, we from cBioPortal database (https://www.cbioportal.org/) to download the TCGA database related clinical information and treatment strategies, conducting clinical data in the download of disease specific survival statistics. Four of these cases had no relevant clinical data despite having gene expression information in the TCGA database. Thus, a total of 174 patients received chemotherapy, endocrine therapy, radiation therapy, and adjuvant therapy, including 102 patients who received gemcitabine. Cox regression analysis and univariate correlation analysis were performed on 174 patients. As shown in Fig. 2A, a statistically significant relationship between PLA2G16 expression and overall survival (OS) in pancreatic cancer patients was confirmed. This analysis encompassed variables such as age, gender, tumor grade, and PLA2G16 expression, revealing that age and PLA2G16 expression stand as independent prognostic indicators (refer to Table 2 and Fig. 2A). Furthermore, patients with pancreatic cancer exhibiting high PLA2G16 expression demonstrated notably shorter OS (Fig. 2B) and disease-free survival (Fig. 2C) periods compared to those with low PLA2G16 expression. Finally, we plotted overall survival among the 102 patients who received gemcitabine (Fig. 2D). Log-rank survival analysis showed that in patients treated with gemcitabine, the overall survival of the PLA2G16 high expression group was significantly lower than that of the low expression group (p < 0.05). Kaplan–Meier survival curves showed that patients with low PLA2G16 expression had a longer survival time, while patients with high PLA2G16 expression had a significantly lower survival rate, suggesting that high PLA2G16 expression may be associated with poor prognosis after gemcitabine treatment. This result suggests that PLA2G16 may be a potential biomarker affecting gemcitabine efficacy or resistance, providing a potential clinical basis for individualized treatment. These results underscore the clinical significance of PLA2G16 as a potential prognostic marker in pancreatic cancer.

Table 1 Clinical characteristics of patients with pancreatic cancer obtained from TCGA database.

Characteristic	Total	Percentage (%)	
n = 178	
Age	
≤70	116	65.2	
>70	62	34.8	
Gender	
Female	80	44.9	
Male	98	55.1	
Race	
White	157	90.2	
Asian	11	6.3	
Black or African American	6	3.5	
T stage	
T1	7	4	
T2	24	13.6	
T3	142	80.7	
T4	3	1.7	
N stage	
N0	49	28.3	
N1, N1b	124	71.7	
M stage	
M0	80	95.2	
M1	4	4.8	
Pathologic stage	
Stage I	21	12	
Stage II	147	84	
Stage III	3	1.7	
Stage IV	4	2.3	

Figure 2 The expression of PLA2G16 strongly correlates with the OS of pancreatic cancer patients.

(A) Multivariate Cox analysis of PLA2G16 expression and other clinical pathological factors, including age, gender, and tumor grade, *: p < 0.05, ***: p < 0.001. (B) Correlation of overall survival rate with PLA2G16 expression in PAAD was analyzed by GEPIA. (C) Correlation of disease-free survival rate with PLA2G16 expression in PAAD was analyzed by GEPIA. (D) Overall survival curves for 102 patients treated with gemcitabine.

Table 2 Univariate Cox regression analysis on disease-specific survival.

Characteristics	Total (n)	Univariate	
	n = 174	Hazard ratio	p value	
Age	
≤70	114	Reference		
>70	60	1.0 (1.01–1.0)	0.015*	
Gender	
Female	80	Reference		
Male	94	1.2 (0.78–1.8)	0.426	
Histologic grade	
Low grade	21	Reference		
High grade	153	1.1 (0.74–1.7)	0.588	
PLA2G16	
Low	87	Reference		
High	87	1.5 (1.23–1.9)	<0.001***	
Note:

* p < 0.05.

*** p < 0.001.

PLA2G16 contributes to gemcitabine resistance in pancreatic cancer cells

To complement our study, we retrieved cell samples from the GEO database, including gemcitabine-sensitive or gemcitabine-resistant cell lines (GSE172303). The downloaded original data package contained the FPKM data processed by the original author, and there were three duplicate data. Subsequently, the data were imported into GraphPad Prism 8.0 for statistical analysis. The analysis of PLA2G16 expression levels shown in Fig. 3A showed that the expression level of PLA2G16 was significantly upregulated in gemcitabine-resistant SW1990 cells (SW1990/GR) compared with gemcitabine-sensitive cells (SW1990). To corroborate this finding, we constructed BxPC-3/GR and PANC-1/GR cells and they were subjected to RNA sequencing. The outcomes confirmed a significant elevation in PLA2G16 expression in the gemcitabine-resistant cells relative to their parental counterparts (Figs. 3B and 3C), thereby suggesting a substantive association of PLA2G16 with gemcitabine resistance in pancreatic cancer.

Figure 3 PLA2G16 contributes to gemcitabine resistance in pancreatic cancer cells.

(A) Differential expression analysis of PLA2G16 in SW1990 and SW1990/GR cells. Data retrieved from the GEO database. (B) Differential expression analysis of PLA2G16 in BxPC-3 and BxPC-3/GR cells. (C) Differential expression analysis of PLA2G16 in PANC-1 and PANC-1/GR cells. (D) Expression of PLA2G16 at protein level in BxPC-3 and PANC-1 drug-resistant cells. (E) IC50 of BxPC-3 and PANC-1 cells to gemcitabine. (F) IC50 of BxPC-3/GR and PANC-1/GR cells to gemcitabine. (G) Knockdown of PLA2G16 using two different siRNAs in PANC-1/GR cell lines, mean ± SD, n = 3. (H) CCK-8 assay depicting the proliferation of PANC-1 cells with or without PLA2G16 knockdown, mean ± SD, n = 3, ****: p < 0.0001. (I) Sensitivity of PANC-1/GR cells with or without PLA2G16 knockdown to gemcitabine, mean ± SD, n = 3. (J and K) Representative images and statistical plots of PANC-1/GR cell clone formation after PLA2G16 knockdown, mean ± SD, n = 3. (L and M) Wound-healing assay of PLA2G16 knockdown cells and quantitative analysis, mean ± SD, n = 3, *: p < 0.05, **: p < 0.01, ***: p < 0.001, ****: p < 0.0001. Y-axis represents the ratio of the scratch width at the detection time point to the initial width.

We then examined the expression levels of PLA2G16 protein in sensitive and resistant cells of two cell lines, BxPC-3 and PANC-1. The results showed that the expression of PLA2G16 was significantly increased in drug-resistant cells (Fig. 3D), further proving that PLA2G16 may be a biomarker of drug resistance, and its high expression can be used to identify drug-resistant populations. Subsequently, we measured the IC50 of gemcitabine in pancreatic cancer cell lines by CCK-8 assay to calculate the resistance index of different cell lines. The results indicate that the IC50 values of BxPC-3 and PANC-1 cells were 1.98 and 1.96 μM, respectively (Fig. 3E). In contrast, the IC50 values of the drug-resistant BxPC-3/GR and PANC-1/GR cells were 52.02 and 83.50 μM, respectively (Fig. 3F). Consequently, the resistance indices were calculated as 26.27 for BxPC-3 and 42.6 for PANC-1. As a resistance index exceeding 15 is indicative of a high level of drug resistance, these findings confirm the successful establishment of drug-resistant cell lines.

To further substantiate the potential impact of PLA2G16 on the proliferation and gemcitabine resistance in pancreatic cancer cells, PLA2G16 knockdown was conducted on PANC-1/GR cells using siRNA-PLA2G16. The effect of knockdown was verified by RT-qPCR, revealing a noteworthy decrease in PLA2G16 expression in the knockdown groups (Fig. 3G). Subsequently, CCK-8 assay showed that after PLA2G16 knockdown, cell proliferation could not be completely inhibited, but slowed down. Due to the small number of initial cells, there was no significant difference in growth and proliferation between day 2 and day 3. However, with the passage of time, the proliferation rate of PLA2G16 knockdown group was significantly lower than that of NC group on Day 4 and Day 5, and the difference was statistically significant (Fig. 3H), suggesting the possible influence of PLA2G16 expression on cell viability in pancreatic cancer cells. Furthermore, PANC-1/GR cells (PLA2G16 knockdown group and the control group) were exposed to different concentrations of gemcitabine and cell viability was assessed after 72 h of treatment. The determination of IC50 values for each group unveiled that PANC-1/GR cells with PLA2G16 knockdown exhibited significantly heightened sensitivity to gemcitabine, with notably lower IC50 values (siPLA2G16-1: 28.52 μM, siPLA2G16-2: 6.817 μM) compared to the control group (85.80 μM) (Fig. 3I). Thus, PLA2G16 plays a key role in the regulation of gemcitabine resistance in pancreatic cancer cells. Additionally, PLA2G16 knockdown reduced colony formation (Fig. 3J) and wound healing (Fig. 3L) in PANC-1/GR cells. After the clone formation assay and scratch assay were repeated three times each, the results were visualized using imageJ and imported into Graphpad Prism8.0 for significance analysis. The results showed that after 10 days of proliferation, the growth and proliferation rate of the knockdown group was significantly lower than that of the control group (Fig. 3K), and the scratch healing rate of the knockdown group was significantly higher than that of the control group at 48 h and 72 h (Fig. 3M). These results collectively suggest that PLA2G16 may be a new therapeutic target for pancreatic cancer.

PLA2G16 is implicated in various biological processes in pancreatic cancer

To further elucidate the pivotal role of PLA2G16 in the management of pancreatic cancer, clinical and gene-expression data from TCGA were utilized, totaling 102 primary tumor samples who underwent gemcitabine treatment. These samples were subsequently categorized into two groups based on median PLA2G16 expression level: a high-expression group (51 cases) and a low-expression group (51 cases). Differential gene expression analysis between these two groups was conducted using the limma package in R software, identifying a total of 119 genes exhibiting differential expression, with 77 genes up-regulated and the remainder down-regulated. The resultant DEGs were visualized using a volcano plot (depicted in Fig. 4A) and a heatmap (showcasing the top 20 genes with the most noticeable up-regulation and down-regulation, as depicted in Fig. 4B). To further delineate the functional implications of PLA2G16 in pancreatic cancer cells, the DAVID database was used for GO and KEGG enrichment analysis of 119 DEGs. The top ten enriched categories, including KEGG pathways such as pancreatic secretion, fat digestion and absorption, protein digestion and absorption, alpha-linolenic acid metabolism, linoleic acid metabolism, arachidonic acid metabolism, glycerolipid metabolism, ether lipid metabolism, bile secretion and the Ras signaling pathway, are presented in Figs. 4C–4F. The significance of these enriched pathways suggests a potential involvement of PLA2G16 in the pathogenesis of pancreatic cancer, possibly through the modulation of lipid-related biological processes. These findings offer valuable insights into the molecular mechanism and potential role of PLA2G16 in the development and progression of pancreatic cancer.

Figure 4 PLA2G16 is implicated in various biological processes in pancreatic cancer.

(A) Volcano plot illustrating RNA-Seq analysis of differential expression of genes in pancreatic cancer related to PLA2G16. Data sourced from TCGA. (B) Heatmap illustrating top 20 differentially expressed genes in pancreatic cancer related to PLA2G16. Data sourced from TCGA. (C) KEGG pathway enrichment of differentially expressed genes in PLA2G16-high and -low expression groups. (D–F) GO_BP, GO_CC and GO_MF enrichment of differentially expressed genes in PLA2G16-high and -low expression groups.

The expression of PLA2G16 correlates with immune cell infiltration in pancreatic cancer

To investigate the relationship between PLA2G16 expression and immune infiltration in pancreatic cancer, the gene expression profiles of the aforementioned 102 samples in each respective group were subjected to analysis using the CIBERSORT computational tool. This facilitated the estimation of the densities of 22 distinct immune cell types in both the high and low PLA2G16 expression cohorts. As illustrated in Fig. 5, T cells CD8, T cells CD4 memory activated, regulatory T cells (Tregs), monocytes, M0 macrophages, macrophages M2 emerged as the primary immune cell populations influenced by PLA2G16 expression in pancreatic cancer. Comparative analysis between the high and low expression cohorts revealed a significant increase in Tregs (p = 0.0091) and macrophages M0 (p = 0.0140) within the PLA2G16 high expression group, while T cells CD8 (p = 0.0250), T cell CD4 memory activated (p = 0.0045), monocytes (p = 0.0054), and macrophages M2 (p = 0.0210) exhibited a decrease in the PLA2G16 high expression cohort compared to the low expression cohort. These findings underscore the potential impact of PLA2G16 expression on immune cell composition in pancreatic cancer.

Figure 5 The expression of PLA2G16 correlates with immune cell infiltration in pancreatic cancer.

The proportion of 22 subpopulations of immune cells was analyzed, with T cells CD8, T cells CD4 memory activated, Tregs, Monocytes, Macrophages M0, and Macrophages M2 being the main immune cells affected by PLA2G16 expression.

Discussion

Recent studies have shown that PLA2G16 is expected to become a new target for tumor treatment. Studies have elucidated its significant roles across various cancer types, including breast cancer and pancreatic cancer (Ding et al., 2020; Xia et al., 2020; Yang et al., 2017), positioning it as a promising candidate for cancer therapy (Yang et al., 2022). Pancreatic cancer, renowned for its formidable and recalcitrant nature, stands among the leading causes of cancer-related mortality (Sung et al., 2021). There exists a critical imperative to foster individualized and targeted therapeutic strategies, notably through the identification of molecular targets capable of enhancing patients’ chemotherapy sensitivity. This inquiry endeavors to examine PLA2G16’s involvement in pancreatic cancer pathogenesis, illuminating its potential dual role as both a prognostic indicator and a therapeutic target.

Elevated PLA2G16 expression has been associated with adverse prognosis and increased metastasis in certain cancers (Liang et al., 2015; Xiong et al., 2014), indicating its potential contribution to cancer progression and aggressiveness. This investigation conducts a comprehensive analysis of PLA2G16 expression within a cohort of pancreatic cancer patients sourced from the TCGA database. Through retrospective examination of clinical RNA sequencing data from 102 histologically confirmed pancreatic cancer patients, our findings elucidate PLA2G16’s tumor-associated expression profile in this population. Multivariate analysis further emphasizes PLA2G16 expression’s role as an independent prognostic determinant for pancreatic cancer patients, thereby highlighting its clinical significance.

Additionally, our study reveals a significant upregulation of PLA2G16 expression in drug-resistant pancreatic cancer cells, with manipulation of PLA2G16 impacting cellular proliferation and sensitization to gemcitabine treatment, ultimately inhibiting cell growth and proliferation. PLA2G16’s influence extends to intracellular signaling pathways such as glycolysis and the MAPK pathway, which are crucial for cancer cell survival (Li et al., 2016; Xia et al., 2020). KEGG pathway and GO enrichment analysis of DEGs suggest PLA2G16’s potential involvement in mediating cellular metabolic pathways in pancreatic cancer.

Furthermore, the interplay between immune modulation and cancer progression, coupled with therapeutic resistance, is well-documented (Bruni, Angell & Galon, 2020; Hinshaw & Shevde, 2019). We found that the high expression of PLA2G16 was not only significantly associated with the alteration of immune cell composition in pancreatic cancer patients, but also suggested its potential immunomodulatory role in the tumor microenvironment (TME), which has not been reported before. The expression of PLA2G16 was significantly correlated with the infiltration level of multiple immune cell types, suggesting that PLA2G16 may affect the antitumor immune response and actively shape the immune landscape of pancreatic cancer. These novel findings provide the first evidence that PLA2G16 is involved in immune regulation of pancreatic cancer, providing a new perspective on its function in TME regulation. PLA2G16 may be a biomarker of chemotherapy response in pancreatic cancer and a promising therapeutic target for immune intervention and personalized treatment strategies. However, the specific mechanism of PLA2G16 involved in immune regulation and chemotherapy resistance of pancreatic cancer needs to be further studied.

Conclusions

In conclusion, our findings implicate PLA2G16 expression in predicting survival outcomes in pancreatic cancer, potentially serving as an independent prognostic factor. The association between PLA2G16 and gemcitabine sensitivity in pancreatic cancer, along with its role in modulating immune infiltration, underscores its significance in disease progression and treatment outcomes. Further research is needed to validate these results and explore the underlying mechanisms of PLA2G16 in pancreatic cancer.

Supplemental Information

Supplemental Information 1 Experimental reagents.

Supplemental Information 2 Experimental instruments.

Supplemental Information 3 Raw images from the GEPIA database and raw images of gene mutation status.

PZF/PZFX files must be opened using GraphPad Prism.

Supplemental Information 4 Raw data of Figure 2, raw pictures of GEPIA, raw data and codes of cox prognostic analysis.

PZF/PZFX files must be opened using GraphPad Prism.

Supplemental Information 5 Raw data from cell experiments as well as processing procedures.

PZF/PZFX files must be opened using GraphPad Prism.

Supplemental Information 6 Raw data from monoclonal experiments and processing procedures.

PZF/PZFX files must be opened using GraphPad Prism.

Supplemental Information 7 ImageJ processing procedure for scratch experiments.

PZF/PZFX files must be opened using GraphPad Prism.

Supplemental Information 8 Raw data of differential genes and analysis codes.

PZF/PZFX files must be opened using GraphPad Prism.

Supplemental Information 9 Raw data of differential genes for Figure 4.

PZF/PZFX files must be opened using GraphPad Prism.

Supplemental Information 10 Raw data for pathway analysis of differential genes.

PZF/PZFX files must be opened using GraphPad Prism.

Supplemental Information 11 Cibersort immune infiltration analysis raw data and codes.

PZF/PZFX files must be opened using GraphPad Prism.

Supplemental Information 12 Cibersort immune infiltration analysis raw data for Figure 5.

PZF/PZFX files must be opened using GraphPad Prism.

Supplemental Information 13 Original data for Figure 3G.

PZF/PZFX files must be opened using GraphPad Prism.

Supplemental Information 14 MIQE checklist.

Additional Information and Declarations

Competing Interests

The authors declare that they have no competing interests.

Author Contributions

Xiaoyu Sun performed the experiments, analyzed the data, prepared figures and/or tables, authored or reviewed drafts of the article, and approved the final draft.

Haiyang Jiang analyzed the data, prepared figures and/or tables, and approved the final draft.

Yufei Huang analyzed the data, prepared figures and/or tables, and approved the final draft.

Jing Xia analyzed the data, authored or reviewed drafts of the article, and approved the final draft.

Jie Gu analyzed the data, authored or reviewed drafts of the article, and approved the final draft.

Xinbing Sui conceived and designed the experiments, authored or reviewed drafts of the article, and approved the final draft.

Xueni Sun conceived and designed the experiments, authored or reviewed drafts of the article, and approved the final draft.

Yucheng Zhou conceived and designed the experiments, authored or reviewed drafts of the article, and approved the final draft.

Data Availability

The following information was supplied regarding data availability:

The raw datasets and codes are available in the Supplemental Files.

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
