# Peer review of "PLA2G16 expression predicts prognosis and gemcitabine sensitivity in patients with pancreatic cancer"

_PeerJ, doi:10.7717/peerj.19517_

## Round 0.1 · original submission · Major Revisions

· Academic Editor

Major Revisions

Please respond in detail to the comments of both expert reviewers

·

Basic reporting

In this manuscript by Xiaoyu Sun et al, entitled "PLA2G16 expression predicts prognosis and gemcitabine sensitivity in patients with pancreatic cancer", the authors present their findings on the potential role of adipocyte phospholipase A2 in pancreatic cancer chemosensitivity and as a prognostic factor. The work is based on database analysis and, to a lesser extent, wet lab work (Figure 3). The manuscript is well written in English, the background and references are sufficient, the figures, tables, and raw data are ok and finally, the results are relevant to the hypothesis of the paper.

Experimental design

The experimental design is appropriate but the methods need to be substantially revised.
1. The authors do not provide the number of replicates/independent experiments/assay they performed
2. The authors do not provide any information on statistics
3. The authors do not provide data on SW1990, BxPC3, how the resistant cell lines were developed and validated.
4. No information on how the images reagrding the colony formation assay were acquired and analyzed.

Validity of the findings

In general, the findings support the conclusions.

Additional comments

Some points that need to be clarified further are discussed below.
1. In the HPA dataset, the authors report 4 out of 10 cases with high or intermediate expression of PLA2G16. However, in the subsequent analyses from the TCGA database, the authors report only high and low cases in exactly the same percentages with half being high and the other half being low expressing samples (both in OS, DFS, and gem resistance analyses). There is no intermediate expression? What is the cut-off?
2. How is disease-specific survival (time) defined in Table 2? There are 174 patients reported in Table 2. What about the missing 4 patients? What was the treatment status of the 178 patients? Did they all receive the same treatment?
3. The data for the cell sampled from the GEO database are not shown. Are the SW1990 analyses laboratory or database analyses? How was the statistical analysis done?
4. In Figure 3E, what do the numbers on the y-axis show? What is also shown in 3E is the growth rate. The authors cannot claim reduced viability as the number of cells at each time point is much higher than the initial population (at time 1d). In this context, they also cannot claim "that PLA2G16 plays an important role in supporting pancreatic cancer cell viability" (line 202).
5. Regarding the colony formation assay, the data provided by the authors are poor. Representative images should have been accompanied by proper graphs with quantitative data and statistics.
6. There is no statistical analysis in Fig. 3I, so the authors cannot claim differences between the different groups.
7. Do the authors have any idea of the real protein levels?
8. The 102 samples used for gem analyses are different than the 178 reported above? If not this means that 76 patients did not receive gem treatment. If this is the case then the previous analysis for OS and DFS must be done separately for the patients who received and those that did not receive treatment. Are there any OS/DFS differences in these 102 gem-treated samples/patients?

·

Basic reporting

Authors have investigated the role of PLA2G16 in the progression of pancreatic cancer and therapy response. More specifically, they have used TCGA datasets to point out the existing correlation between expression and poor prognosis; and evaluated the relationship between PLA2G16 expression and various biological processes such as lipid metabolism and immune infiltration using computational tools. Further siRNA screenings demonstrated that downregulation of PLA2G16 re-sensitizes a gemcitabine resistant cell line to treatment.

Although conclusions are well supported by the data and the manuscript is well written, clear and concise, some important information regarding CCK-8 assays and statistical analysis are missed in material and methods. I would also suggest the authors to provide the sensitivity index in figure 3.

Experimental design

The research question is well defined although not novel. The experimental design is limited which makes difficult to draw conclusions regarding the therapeutic potential of PLA2G16. For instance, no tox studies are shown to demonstrate therapeutic window.

Validity of the findings

Most importantly and despite the interesting of this study, similar results were already published in 2020 by Wei Xia et.al. on an article describing PLA2G16 as a mutant p53/KLF5 transcriptional target in pancreatic cancer. Actually, similar experimental design and figures are presented in both articles which makes me question the novelty and relevance of this new study. For this reason, I would suggest the authors to deep in their investigation so to provide new evidence regarding the feasibility of using PLA2G16 as a therapeutic target and prognostic marker.

Additional comments

Just for further reflection and in views of the low PLA2G16 mutation frequency (0.3%) in pancreatic cancer, I believe considering PLA2G16 as a candidate for cancer therapy might present some challenges due to the low number of patients who could probably benefit from it. Therefore, further benefit-risk assessments and clinical validation as a prognostic marker would definitely be of interest for the scientific community.

---

## Round 0.2 · Minor Revisions

· Academic Editor

Minor Revisions

Dear authors, thank you for your revision. However, some aspects require clarification before acceptance:

- where did the BXPC-3 cells came from? their culture conditions?

- gemcitabine at 1/4 IC50 equals what? (actual concentration!), how often was the media changed? what passages of cells were used? you say " the medium was supplemented with drugs every 48h" (ln 209-210), drugs and their concetration exactly? How is it that "Thereafter, the concentration of gemcitabine was gradually increased every 1-2 weeks until it reached 50 µM, and the drug-resistant cells were continuously cultured. ", continuously cultured for? What CCK-8 assay did you actually performed? commercial? ref. or catalogue #, company, etc?

- Then you say that the IC50 was determined? ln -213 - how ? based on...

- PANC-1 cells were harvested from what? "seeded in 96-well plates containing cell suspensions of 5000 cells per well. " - what cells suspensions? from what?
(thus far i think it may be an issue of completeness and language)

- "After 24 hours of incubation, the cells were treated according to the experimental conditions. After 72 hours of drug treatment"(continues in subtitle 2.6 line 214 on...), - according to what experimental conditions? what does this mean? what exactly was the protocol for the drug treatment? ln 219 cells incubated for 1 or 2h? (1h difference can be relevant!), incubator with which settings? 45ºC? model and brand and city/country of the microplate reader?

- ln 223 "treated differently according to experimental requirements" - how, what, when exactly was treated with what? what proteins were extracted "for WB"? - total protein?

- this goes on and on... requires completeness! details! and do not forget reagents data (including kits' numbers etc). For SDS-PAGE, what type of gel did you use? how much protein did you load? etc etc ... what antibodies were used? concentrations?

- same issues with the qPCR methods description... and transfection...

- what cells were used for the colony formation assays (remember you mentioned 2 cell lines before!!)... what fixative was used? how long? concentration.... did you use any patch with imageJ for the counting? etc etc... methods need absolute revision for completeness!

·

Basic reporting

-

Experimental design

-

Validity of the findings

-

Additional comments

The authors have properly addressed all my previous concerns. I have no further comments.

---

## Round 0.3 · accepted · Accept

· Academic Editor

Accept

Dear authors,

thank you for your revisions and endeavours! I am now moving forward with your manuscript. Congratulations! Please, be thorough during production stages with your proofreading (I think at least some figures do not reach enough amplification and resolution / quality) and thank you once again.